# The Objectives of Farm Operations—Evidence from a Region in Poland

**Aleksander Grzelak**

Department of Macroeconomics and Agricultural Economics, Institute of Economics, Poznań University of Economics and Business, 61-875 Poznań, Poland; agrzelak@interia.pl; Tel.: +48-88888-7661

**Abstract:** The objectives set by agricultural producers directly translate into the directions of the development of farms. What is particularly important: Is there a conflict between the economic and environmental objectives of farm operations? This issue is not resolved in the literature on the subject and still is a challenge for policymakers. The main aim of the article is to identify the preferences of farm managers concerning the objectives of farm operations, as well as to examine their mutual relations. The article employs the results of surveys carried out on farms from the Wielkopolska region (Poland). The analysis of Spearman's rank, cluster analysis, also the classification tree method, and multidimensional scaling were applied. The research shows that the relations between the income and assets objective turned out to be moderate in terms of strength, while the environmental objective turned out to be statistically insignificant related to economic objectives (in the context of their perception by respondents). There are differences in this respect, including also a group of the respondents in which income and environmental objectives have been ranked simultaneously high. It is, however, difficult to state clearly whether there is a complementarity between the income and the environmental objective from the perspective of perception by respondents. Although when the context of the real action is taken into account then the answer should be positive. Therefore, there is a gap between the farmers' perception of reality—choice of the hierarchy of objectives, and the real activities, e.g., in terms of pro-environmental activities. It is a new issue that points to the need to stimulate the environmental objective, in particular, through support at the level of agricultural policy instruments.

**Keywords:** farms; objectives; income; assets; the environment; the Wielkopolska region

## 1. Introduction

The research on farm objectives refers to both the classical economic theory related to the microeconomic producer theory [1] and heterodox currents such as behavioral economics or complexity economics [2]. This is due to both the economic and psychological context—issue of perception of the objectives of these problems. Such an approach, also called pluralistic, can be found particularly in the case of sustainable development issues [3]. Agricultural producers are guided by different objectives in meeting their needs for both the operation of the farm and the household. This is the basis for achieving success and satisfying needs. Knowing the objectives of farms allows for changing the preferences of farmers, the strategy of their actions, as well as the possible acceptability of actions in the sphere of agricultural policy and rural areas. In classical terms, the main objective of an agricultural producer is to maximize income [4,5]. If we take into account the combination of production and consumption functions, the perspective of perceiving these phenomena is broadened and complicated. Then, the maximization of income may be a means to satisfy consumption needs, aspirations related to persons forming a household, e.g., obtaining appropriate education, or generational ones related to ensuring the continuity of the functioning of an agricultural holding (transfer to a successor). As studies

stress [6], farmers do view goals in a multidimensional framework. The recognition of preferences in the scope of the objectives of farm functioning is important for shaping the agricultural policy at the level of the EU (European Union) and its member states. It concerns economic issues, but also environmental and social issues.

The article aims to recognize the preferences of farm managers in the scope of the objectives of the farms' operations, and also to examine the mutual relations between those objectives. The article focuses more on the most important (according to respondents) economic objectives: "Providing income" and "increasing the value of assets", as well as the environmental objective. The importance of the latter results from the more and more widely exposed environmental dimension of agricultural activity, both at the institutional level (EU—the CAP (Common Agricultural Policy) instruments) and at the level of the farms themselves. The point is the necessity of environmental adjustments, an increase of awareness of agricultural producers also as consumers. One of the contemporary challenges for the development of agriculture is to reconcile economic and environmental objectives. Moreover, the implementation of objectives relating to the farmer's family is connected with economic objectives. Without the latter, it would be difficult to meet generational objectives when in the examined group the average share of income from agriculture in the total household income of a farmer's family is dominated and was 76%. Therefore, because the majority of the respondents' income comes from agriculture, the income objective of the functioning of an agricultural holding is so important in the hierarchy of their objectives. Two research hypotheses were made: (1) There is a strong link between the choice of income and asset objectives by respondents and (2) there is not a clear, but positive link between the economic objectives (providing income, increasing the value of assets) and the environmental objective.

The issues raised are not unequivocally resolved in the literature. It is particularly important whether there is a conflict between the economic and environmental objectives on farms [7,8]. It is also a question of whether larger units have a better chance of sustainable development and thus there is a positive relationship between the economic and environmental objectives [9,10] or a negative one [11]. The objectives set by agricultural producers directly translate into the directions of the development of agricultural holdings.

The themes relating to the objectives of farm operations also have an application dimension. Therefore, it is important that in the next EU budget perspective and the anticipated changes in the CAP, support instruments should be adjusted to stimulate sustainable development taking into account both the economic objectives of agricultural producers and the environmental objectives. It should also be noted that the farmers' perceptions of the hierarchy of the objectives of farm operations may not be adequate for the activities they undertake. Therefore, the recognition of these processes may contribute to more effective implementation of pro-environmental measures by farmers (problems of economic incentives, education, improvement of awareness). Analyzed issues are also important from a climate change perspective and the challenges of the food sustainability policies [12]. Contribution to the existing discussion about the objective of farm operation was conducted in two ways. Firstly, to recognize them as well as their mutual relations on the example of the region in an EU country with a medium level of development (Poland). Secondly, the research results provide arguments for further strengthening of the environmental component of the policy (CAP) as well as for a holistic approach to agriculture and rural areas. In addition, it can be seen that surveys of farm operations were undertaken several decades ago more often than today. Due to the dynamic geopolitical environment, there is a need for further research in this area.

In the Introduction, the motivation to conduct the research, and the hypotheses were presented; in the Literature Review, the current state-of-the-art on the raised issues; then, the methodology was used. Afterward, the obtained research results are analyzed, which is compared with other outcomes, and at the end of the article, the conclusions, reflections, and the implications for policy adjustments are presented.

## 2. Literature Review

Due to the family nature of farms, there are different priorities of household members, which makes setting goals a complex but also dynamic process [13,14]. This is related both to the economic (wealth level) and behavioral dimension relating to the subjective norms of farm users, their needs and aspirations, risk propensity, or attitude towards the future. There are many different possibilities for classifying agricultural objectives. The literature often mentions bundles of objectives [15,16]. For example, based on the research of Kallas et al. [17], the Catalonian vineyard farmers distinguish between economic, environmental, and socio-cultural objectives. In turn, Majewski and Ziętara [18] indicated the following hierarchy of objectives for individual farmers based on a survey conducted in Poland of 655 farmers: 1. Raising children and ensuring a good future for them; 2. certainty of selling the products; 3. certainty of keeping the farm free from debt and risk; 4. maximum income from the farm; 5. modernization of the farm. Only in the fourth position the question of maximizing income from agriculture arises. This distribution could result from unstable farming conditions in agriculture in Poland in the 1990s. Sulewski's research [14] carried out on agricultural holdings in Poland shows that farmers indicated the highest preferences for "increase in income from the holding", "raising children", and "ensuring their good future" in the hierarchy of objectives. The last of the above-mentioned objectives concerned farmers who have children. In turn, Khan and Chander [19], based on the results of research carried out on cattle and buffalo farmers in India, indicated that the most important objectives were: a certain and stable income, inherited business, and, in fourth place, maximized profit.

By achieving their objectives, farms are in multifunctional entities. The hierarchy of objectives of the functioning of agricultural producers is determined by the phase of the life cycle of an agricultural holding, determined by the age of the head of the holding, having a successor, and a multi-generational family. Generational changes in farm management constitute a natural development mechanism. Therefore, having a successor favors the modernization of a holding and increases the importance of objectives directly related to the functioning of an agricultural holding. In general development processes, conditions relating to family issues, including the need to educate the young generation, as well as the relationship between income and work, increase the pressure on the increase of benefits of farming for the household. This means an increase in the importance of household objectives. Already in studies from the 1960s, Dorenkamp [20], on the example of German farms, pointed to the decreasing role of income for the sake of stability ("quiet life"). In turn, Willock et al. [21] emphasized the role of financial, socioeconomic, and psychological factors in determining the farmers' behavior and objectives of household functioning.

Nowadays, the environmental dimension is an important element in the assessment of agriculture and farms. This is due to the very changes made in the CAP towards the environment, the increased social pressure associated with it, and the challenges of climate change. To achieve the environmental objective more effectively at the level of the agricultural producer, economic incentives are needed that address these issues and allow for the valorization of public goods or environmental services provided by agricultural holdings. An open question is still unresolved: Is it possible to achieve economic and environmental objectives simultaneously? For example, Dolman et al. [22] investigating economic, environmental, and societal performance among Dutch fattening pig farms pointed out that some agricultural entities outperform others parallel in economic, environmental, and societal fields. Similar conclusions can be found in the article by De Koeijer et al. [7]. According to their study, the Dutch sugar beet growers achieve economic and environmental efficiency at the same time. In addition, there is a potential to improve the results without conflicts between the economic and environmental goals. Ryan et al. [23] proved that the top-performing dairy farms (in economic terms measured using the productivity of production factors, and farm viability) tended to be the best-performing farms from an environmental sustainability perspective. The point was the lowest greenhouse gas emissions per unit of product. Moreover, the outcomes Villalba et al. [24] conducted in sheep farms in the Basque Country (Spain) are in line with this and they show a complementarity between the economic and

environmental performance (nitrogen excretion). This would mean that economic and environmental objectives could be reconciled. When assessing the hierarchy of objectives, it is also important to be aware of the behavioral context, resulting from the axiological attitudes of agricultural producers' different needs [25]. Hence, managers of farms strive for both economic rationality (income objective) and family rationality (objectives relating to the farmer's family). As it results from the research by Pennings and Leuthold [26], the behavioral characteristics and objectives, which are implemented by farmers and their market orientation, play important roles in the development of farms.

## 3. Materials and Methods

As a data source, the results of surveys carried out in January and February 2020 was used on a group of 120 farms from the Wielkopolska region (Poland), which are a part of the farm accountancy data network (FADN). The research tool was an interview questionnaire entitled: "Assets and income in agricultural holdings in the paradigm of sustainable development" (The questionnaire S1). The questionnaire was divided into three sections: General and economic-financial, environmental, and other questions . The following research scheme was used [27,28]:

- Step one: The selection of analyzed scientific problems.
- Step two: Identification of the number of respondents to be interviewed—managers of farms in the Wielkopolska region. In this case, the aim of maximizing the cost-benefit effects was dominated. The number of 120 was considered as relevant to the level of the raised issues and adequate based on the author's judgment [28].
- Step three: The selection of holdings. The research sample was based on the economic size (ES) of the farms. The economic size class is defined as the sum of the standard value of agricultural output, the so-called Standard Output (SO). It is the (SO), the average monetary value of the agricultural output at a farm-gate price of each agricultural product—crop or livestock in a given region, and is expressed in thousands of EUR. The analyses used the delimitation of six classes of economic sizes: Very small farms ES1 (size 2–8 thousand EUR SO), small ES2 (8–25 thousand EUR SO), medium ES3 (25–50 thousand EUR SO), medium-large ES4 (50–100 thousand EUR SO), large ES5 (100–500 thousand EUR SO), very large ES6 (over 500 thousand EUR SO). Moreover, the production types of farms have been taken into account classification (TF—type of farming) into eight groups of farms. It is the system for distinguishing eight types of production of agricultural holdings within the framework of the EU FADN agricultural accounting according to the predominant production direction. A quota selection of the number of farms for the survey was applied. For this purpose, the assumed number of the surveyed farms (120) was divided proportionally taking into account both the economic size (ES2–ES5) and production type of the farms (TF1—fieldcrops, TF5—milk, TF6—other grazing livestock, TF6—granivores, TF8—mixed), which occurs in the group of farms conducting agricultural accounting according to FADN in the Wielkopolska region [29]. Next, the farms were drawn using the operators of the Agricultural Accounting Office of the Institute of Agricultural and Food Economics in Warsaw.
- Step four: Questionnaire preparation and validation [28]. The final version of the survey was verified by scientists experts and practitioners The selected questions of the questionnaire, used in the article, are in the supplementary materials.
- Step five: Conducting surveys. The interviewers (35) were allocated to the selected farms. They were advisors of the Agricultural Advisory Centre dealing with agricultural accounting in these drawn farms, including care of correct entries in the farm's accounting book. This made it possible to obtain research material of high reliability. Only in a few cases (9) did the questionnaires require supplementing or individual explanations from the interviewers, also in situations of so-called outlier observations.

With this type of research, it is important to test for the overall score reliability. The most often assessed issue is the internal consistency reliability, which refers to the degree to which responses

(in this case the validity of objectives of farm operations) are consistent across the items within a measure. To measure this issue, the value of Cronbach's alpha is reported. The items included in the analyses $\alpha = 0.78$, which means that it exceeds the suggested threshold of 0.5 and indicates a high reliability of the position of the scales for the examined purposes [30].

In the study, the correlation of Spearman's rank, the multidimensional scaling, cluster analysis, and classification trees were employed. The second of these was used only to reduce the data. The multidimensional scaling does not require any assumptions regarding variable distributions. The computational method of multidimensional scaling involves minimization of the function called stress function or slightly modified standard stress function: Coefficient of alienation on the basis of matching the configuration of the distance measuring points [31,32]. The stress function has been written as the following Formula (1):

$$S_{(d_{ij})} = \sum \left( d_{ij} - f\left( \delta_{ij} \right) \right)^2 \tag{1}$$

where $d_{ij}$ is the reproduced distance at a given number of coordinates in the space of scaling, $f(\delta_{ij})$ is the monotonic function of initial distances.

The lower the stress value, the better the match between the reconstructed distance matrix and the observed distance matrix. For example, Bogarti [33] suggests that two- or three-dimensions are suitable for data presentation if the STRESS value for a given number of dimension is below 0.15.

The article also uses cluster analysis, which is used in research in the food economy as well as in sustainability issues [34]. In the case of cluster analysis, the Euclidean distances were employed and Ward's agglomeration method, which is based on variance (minimizing the sum of squares of deviations of the objects inside clusters) [35]. This made it possible to separate groups of respondents (managers of farms) due to the similar preferences in terms of the objectives of farms' functioning. The number of clusters was determined based on the diagram of the tying distance in relation to the stages of tying and the assumption of the minimum number of units in the cluster, i.e., 15, which in consequence made it possible to separate five clusters. Variables used in the cluster study were tested with the analysis of variance to assess their usefulness for differentiation of the groups. It turned out that all analyzed variables significantly differentiate the listed five clusters at the level of 0.05. Due to the fact that cluster disconnection is an important element of these analyses, Levene's test was also conducted. The hypothesis of homogeneity of variance between the distinguished clusters at 0.05 was rejected, except for the variable "modernization of farm". At the same time, the Mann-Whitney U test to assess the significance of differences between medians from tested clusters, confirmed that clusters differ significantly in the preferences of respondents in terms of the hierarchy of the objectives of farm operations. Only in the case of one objective (out of eight)—"providing of income", there was no significance of differences between clusters, because this objective was widely the highest ranked among respondents.

In turn, the classifications trees were applied to the objective which obtained the top rating among the respondents—"providing income". This method makes it possible to choose by respondents the highest preference for this objective. This is important for the knowledge of the development mechanisms of agricultural holdings. The applied classification tree method is based on the C&RT (Classification and Regression Trees) algorithm and was promoted by Breiman et al. [36]. The main idea of this method is to find rules in the form of a set of logical division conditions of the type "if...that...". This makes it possible to classify objects by building a model-tree. The main advantages of this method include non-linearity and non-parametric. Therefore, the relationships between variables do not have to have a normal distribution and, moreover, there is no need to standardize the variables of differentiation. It is not without significance that there is no need to make assumptions about the nature of the relationship between predictors and the dependent variable, the possibility of classifying incomplete data, using the same variables in different parts of the tree, taking into account both quantitative and qualitative variables, the transparency of this method (graphical presentation),

simplicity, and allowing easy classification of new cases, as well as clarification of its rules. In the case of defects of this method, however, it happens that the models proposed by classification trees are complex and difficult to interpret. This is most often due to the suggested division criteria assigned to a particular node, which are not confirmed by economic reasons. Bias in the split rule selection [37] or limited stability is also indicated. Therefore, various alternative proposals have been made to improve the traditional approach. These include random forests [38], bootstrap bumping [39], QUEST [40], or the TARGET genetic algorithm approach [37]. Each of these methods has its own research limitations. However, as these analyses are not about the classification accuracy of trees, but more about diagnosing of the factors that better discriminated for the highest ranked income objective as an alternative to regression analysis, the traditional C&RT approach has been used. This algorithm is relatively commonly used and its effectiveness is also repeatedly stressed against the background of alternative decision tree algorithms [41–43].

Finally, the quality of the matching of the received rules was verified by a classification matrix from which it was possible to identify cases properly and incorrectly classified by the model. Moreover, it was assumed that the quality control of the model will be performed with the use of a V-fold cross-check, assuming V = 10. This meant a random extraction of 10 subsamples from the examined observations. Each subsample is used nine times (V−1) in the teaching sample and one time in the test sample. For each test sample, the costs of the cross-check are calculated, which are then averaged. The idea is to select the smallest tree whose cross-check costs will be no more than the smallest (in the whole tree sequence) cross-check costs plus one standard deviation of these costs [44]. In such a situation, all end nodes are relatively homogenous and of a low number. The choice of the optimal tree is a compromise between the complexity of the tree and the accuracy of the mapping expressed by the costs of the cross-check and resubstitution. The cost of resubstitution is calculated based on the proportion of cases misclassified by a classification model built based on all cases. The principle of one standard deviation is applied, which allows the identification of the smallest tree size, whose cross-check costs differ a little from the minimum test costs for a sequence of all the trees analyzed. The difference between the tree with the lowest costs and the optimal tree should be less than one standard deviation (2):

$$SK \leq \min (SK) + \sigma \min (SK) \tag{2}$$

where SK—costs of the cross-check for the classification tree under analysis, min (SK)—minimum costs of the cross-check for the sequence of trees analyzed, and σ min (SK)—standard deviation of the minimum costs of the cross-check.

Moreover, it was assumed that the interruption of the process of creating new tree nodes is realized by using pruning according to the criterion of the minimum number of observations in the split node (at n ≥ 15). The Gini measure [45] was used to select the best division.

In the case of the conducted surveys, the classification trees concerned the classification of farm managers who rated the objective of "providing income" at the highest on a 5-grade scale. In the course of further analysis, modifications were made to the selection of this objective for researches. The point is that the group of farm managers indicating the highest preferences for the objective "providing income" included almost everyone (about 90%) who also chose the second (in terms of hierarchy) economic objective "increasing the value of assets" at the highest level (in the 5th scale). Therefore, one was decided that in the analysis of the income objective related to classification trees, the units which gave the highest preference will be subject to the researches, while simultaneously the asset objective did not receive the highest preferences. Therefore, 55 cases were included in the analysis of classification trees because 29 observations were excluded. This exclusion made it possible to distinguish more clearly rules for selecting the highest preferences for the income objective. The starting point for the application of this research method was the selection of variables. Their selection was based on the criterion of relevance for the studied variables, the logic of mutual relations, taking into account the economic as well as environmental context. A wide range of explanatory variables was initially qualified because the complexity of the phenomenon under investigation is difficult to present with just

a few of them. Then, in the course of the model building, those variables were eliminated whose value in the importance ranking (according to the C&RT algorithm) was at the level of 10 and less (100 was the maximum value) (Table S1). The aim was to distinguish more clearly the factors shaping the most preferred objective by respondents: "Providing of income" in the form of rules during further analysis.

## 4. Results

Wielkopolska is one of sixteen voivodships (regions) in Poland. The utilized agricultural area is 11.3% of all in Poland and the value of agricultural gross output is 17.4% of all in Poland. The surveyed group of farms are comparable to farms from some countries of Southern Europe (Portugal, Italy) and more favorable than in Greece. However, the structure of the production is different in these countries because the production of fruit and vegetables is dominant. In the case of the Wielkopolska region, the production of pigs is the most important, although farms without specialization dominate in the sample (Table 1). The average area of the utilized agricultural area in the analyzed group of farms in the Wielkopolska region was higher than the average farm covered by the FADN system in Poland (29 ha in the studied farms compared to 20.5 ha in the FADN in Poland), the value of total assets was 1.8 times higher in the surveyed units, investments were 2.4 times higher, and the income was 2.2 times higher. Hence, the surveyed holdings achieve more favorable results than the average farm in Poland covered by the FADN system. Selected descriptive statistics are set out in Table 1.

**Table 1.** Selected descriptive statistics of the farms surveyed in the Wielkopolska region (2018).

| Specification | Mean | Min. | Max. | SD * |
|---|---|---|---|---|
| the utilized agricultural area (ha) | 28.99 | 4.22 | 151.15 | 22.77 |
| share of agricultural income in total household income (%) | 76.24 | 10.00 | 100.00 | 27.46 |
| share of plant cover on arable land during winter (%) | 49.63 | 0.00 | 100.00 | 26.45 |
| stocking density (LU/ha) | 1.36 | 0.00 | 10.41 | 1.39 |
| agriculture income (thousand EUR) | 19.25 | -3.35 | 102.58 | 90.90 |
| value of agricultural output (thousand EUR) | 72.41 | 5.39 | 412.80 | 233.05 |
| total liabilities (thousand EUR) | 19.15 | 0.00 | 443.93 | 242.66 |
| value of assets (thousand EUR) | 327.72 | 30.30 | 1351.30 | 1038.81 |
| share of farms with productive orientation (%): | | | | |
| fieldcrops | | | 23.3 | |
| milk | | | 16.7 | |
| other grazing livestock | | | 9.2 | |
| granivores | | | 11.7 | |
| mixed | | | 39.1 | |
| share of farms taking into account economic size (p. 4) (%) | | | | |
| ES2 | | | 25.8 | |
| ES3 | | | 29.2 | |
| ES4 | | | 25.8 | |
| ES5 | | | 19.2 | |

N = 120, * SD—standard deviation. Source: Own study based on the questionnaire survey.

According to the conducted farm surveys in Wielkopolska, farmers assessed "providing income" as the highest in the hierarchy of objectives: (4.55 out of 5), and "providing resources for the family" (4.48 out of 5) (Table 2).

**Table 2.** Independent samples tests in relation to the assessment of the importance of the objectives of operations of the examined agricultural holdings in the Wielkopolska region (list of the most important economic objectives: "Providing of income" with the other objectives).

| | Specification | Mean for the Objective "Providing of Income" (1) | Mean for the Other Objectives (2) | *t* * | df | *p* |
|---|---|---|---|---|---|---|
| | | **list of the objective "providing of income" (1) vs. the other objectives (2)** | | | | |
| (1) | (2) | (1) | (2) | | | |
| | "increasing the value of assets" | 4.55 | 3.78 | 7.35 | 238 | 0.00 |
| | "providing funds for the family" | 4.55 | 4.48 | 0.66 | 238 | 0.51 |
| "providing | "modernization of farm" | 4.55 | 3.53 | 9.24 | 238 | 0.00 |
| of income" | "ensuring continuity of running" | 4.55 | 3.88 | 6.44 | 238 | 0.00 |
| vs. | "care for the environment" | 4.55 | 3.82 | 7.16 | 238 | 0.00 |
| | "increase of output" | 4.55 | 3.52 | 10.30 | 238 | 0.00 |
| | "stabilization of the economic situation of the family" | 4.55 | 4.38 | 1.79 | 238 | 0.07 |

N = 120; the hierarchy of each objective from 1 to 5; *t* *—Student's *t* test; df—degrees of freedom; *p*—level of significance for the test. Source: Own study based on the questionnaire survey.

The objective of "increasing the value of assets" was rated slightly below the average of the specified objectives (sixth out of eight). This objective was statistically differed in plus only from two objectives ("increase in production", "modernization of farms"), while in the case of three objectives in minus. In turn, the objective "care for the environment", was rated slightly higher (fifth place) than the "increase in the value of assets" objective, although the difference between them was not statistically significant. This means that environmental issues in the group of respondents were quite important and certainly not marginal. This is due to the growing importance of the environmental dimension in CAP EU support instruments. On the other hand, there is also a growing awareness of farmers of the increasing importance for consumers of environmental issues. The greatest importance for the income objective is also related to the growing commercialization of rural activities. Moreover, not without significance is the context of the political rent, which makes part of the income easier to receive and dependent not on the improvement in farming efficiency or increase in resources, but on entrepreneurship in obtaining support funds.

It is difficult to draw a clear conclusion about the objective "increasing the value of assets" in agricultural holding. In light of the conducted studies, it is not of primary importance but, on the other hand, it is not marginal and is of higher importance than the objective of the "modernization of farm". Resources may be treated by farmers with less priority than streams (income), which accumulate after time (by a propensity to invest) in resources (capital, land). Therefore, they are secondary to income. On the other hand, the resources allow for the creation of income streams. An agricultural producer is faced with a choice: To maximize the current income, or to invest in assets in order to increase future income. In light of the presented research results, the preference is rather for the first solution. In the case of the latter, it would mean the so-called investment approach. However, the analysis of data shows that this was not the case. There were higher resources (land and capital), a higher level of income, investments also in relation to income and equity, and liabilities in the case of respondents who prefer the income objective the most.

The results of multidimensional scaling confirm previous conclusions. A two-dimensional model was adopted for research, which results from a very low stress level (0.035) and alienation coefficient (0.056), which indicates a good matching of the model to empirical data. It was observed that the objective "care for the environment" was different from the other, especially from "providing of income" (Figure S1 and Table S2). Moreover, the objective "increasing the value of assets" is somewhat out of step with this approach. This would mean that the respondents' perception of the goals is multidimensional with a shorter time horizon for the income objective than for the asset one. In turn,

the objective "providing of income" and "providing funds for the family" are closest to each other, which results from the combination of the functions of a farm and a home in a family work situation. The point is also that these objectives have recorded relatively high values of the first dimension with a low second dimension, which means a relatively homogeneous perception of these goals (Figure S1) by respondents.

The factor influencing the functioning of agricultural holdings is the scale of production and the related level of income in relation to the parity income, or the share of income from agriculture in the total household income. In a situation of the decreasing importance of agricultural income, the importance of objectives relating to the household increases. The use of one's own labor resources on the farm is also of particular importance here. This determines the family structure of the holding, or the similarity of the holding to an enterprise when paid labor dominates. In such a situation, maximizing income or even profit is the most important in the hierarchy of objectives of the agricultural holding.

In the case of the three objectives, including the "providing of income", belonging to a specific group of economic size of farms, significantly influenced the differences in respondents' preferences for the hierarchy of selection at the level $p < 0.05$ (Table 3). In the case of the classification by production types of farms, the differences for all objectives proved to be statistically insignificant. For none of the objectives, $p$ was less than 0.1 for the non-parametric Kruskal-Wallis test. It is worth pointing out, however, that the problem of the importance of specialisation for economic and ecological trade-offs of agricultural specialization has been addressed, for example by Klasen et. al. [8]). Interestingly, in the case of the environmental objective, differences between groups of holdings due to economic size proved to be the least statistically significant. In larger units, the income objective, also the asset objective (although to a lesser extent) was rated even higher by respondents than average (Table 3). On the other hand, however, as it results from these surveys, larger farms more often took pro-environmental measures. 73% of the surveyed households that took at least 6 pro-environmental measures (out of 15 specified in the survey: 1. reducing the use of plant protection products per ha; 2. reducing fertilizer use per ha; 3. reducing stocking density (per 1 ha of UAA); 4. using catch crops; 5. plowing straw on arable land; 6. afforestation of land; 7. increasing the share of permanent pasture; 8. set-aside; 9. reducing the share of cereals in the sowing structure; 10. using an arable land cover with vegetation during winter; 11. thermo-modernization of buildings; 12. replacement of traditional septic tanks for ecological purposes (or sewerage connection); 13. changing the heating furnace (building, utility room) to a more modern one; 14. modernization of plant protection products (fertilizers) storage place; 15. Other (which ones?). In the last one the respondents most often reported liming of soil 5-cases) in the period 2016–2019 belonged to the ES4–5 group.

**Table 3.** The validity of the objectives of operation in the examined agricultural holdings in the Wielkopolska region due to the economic size of farms.

| Objectives | Economic Size of Farms (ES) | | | | $p$ * |
|---|---|---|---|---|---|
| | ES2 N = 31 | ES3 N = 35 | ES4 N = 31 | ES5 N = 23 | |
| "providing of income" | 4.35 | 4.63 | 4.42 | 4.87 | 0.03 |
| "increasing the value of assets" | 3.71 | 3.80 | 3.71 | 3.91 | 0.77 |
| "providing funds for the family" | 4.10 | 4.51 | 4.68 | 4.70 | 0.06 |
| "modernization of farm" | 3.13 | 3.26 | 3.84 | 4.04 | 0.00 |
| "ensuring continuity of running" | 3.61 | 3.89 | 3.97 | 4.09 | 0.11 |
| "care for the environment" | 3.87 | 3.77 | 3.81 | 3.83 | 0.94 |
| "increase of output" | 3.19 | 3.43 | 3.61 | 3.96 | 0.01 |
| "stabilization of the economic situation of the family" | 4.23 | 4.29 | 4.45 | 4.61 | 0.30 |

N = 120; the hierarchy of each objective from 1 to 5; $p$ *—refers to the significance of non-parametric test of the Kruskal-Wallis for the significance of differences for many independent groups. Source: Own study based on the questionnaire survey.

At this point, it should be stressed that, according to the cluster analyses (Figure S2 and Table S3), the surveyed group of respondents is not homogeneous in terms of preferences to the hierarchy of objectives. In one of the five clusters (no. 5), the maximum preference for the objective "providing of income", high for the objective "increasing the value of assets" (4.4), and the highest (compared to other clusters) for the environmental objective (4.6) could be noted. The units of this cluster were characterized by a higher average area of agricultural land (32 and 30 ha in others), higher value of assets (10%), agricultural output (27%), level of investments (34%) compared to others. At the same time, there was a higher share of cover of arable land with catch crops (21%) and cover during winter (7%), as well as a rotation of sowings was applied in all units. This may explain the previous contradiction regarding the not very high positioning of the environmental objective on average in the whole study group.

It is also important to note that the significance of the asset, despite the increase in the average assessment, and environmental objective relatively decreased in comparison with the other objectives in the group of the largest agricultural holdings. This resulted from greater preferences for the objective of "increasing output" and "modernization of an agricultural holding". The managers of these farms gave higher preferences for all objectives, which resulted from the increased importance of agricultural income in farmers' household income and thus the function of the farm. Thus, managers of economically stronger farms preferred economic objectives in relation to the environment, despite the fact that they implemented environmentally friendly actions to a greater extent. This can be seen as a kind of compensation for the increased environmental pressure exerted by these units. However, we should be aware of the fact that respondents are different in terms of the hierarchy of objectives, and the delimitation of relatively homogeneous groups is complex and takes into account not only the size of farms. According to studies by Westbury et al. [46], the bigger the farm, the better the environmental performance (measured by the Agri-Environmental Footprint Index). They point out that large farms use land less intensively, and provide greater proportions of low input habitats increasing the values of land use diversity, while small livestock farms use more energy and water per ha of the utilizable agricultural area compared to the large farms. Gomez-Limon and Sanchez-Fernandez [9], justifying why larger farms benefit in the environmental field, pointed out: Better implement techniques (minimizing cost soil cultivation, direct sowing), and greater opportunities to participate in agro-environmental programs because of lower transaction costs and better adjustment to the requirements. It was noticed that the respondents who knew their successor were a little bit higher in their assessment of almost all objectives, which indicate that the functioning of the farm is then relatively more important for them (Table 4). More clear differences were noted for economic objectives, especially for the objective "increasing the value of assets", "increasing the value of assets", and "providing of income".

**Table 4.** The preferences of the managers of the surveyed agricultural holdings for the objectives of operations of the agricultural holdings due to the existence or not of a successor.

| Objective | The Successor Is Unknown | The Successor Is Known | *p* * |
|---|---|---|---|
| "providing of income" | 4.45 | 4.62 | 0.29 |
| "increasing the value of assets" | 3.59 | 3.86 | 0.04 |
| "providing funds for the family" | 4.39 | 4.57 | 0.76 |
| "modernization of farm" | 3.46 | 3.55 | 0.33 |
| "ensuring continuity of running" | 3.80 | 3.98 | 0.83 |
| "care for the environment" | 3.84 | 3.60 | 0.08 |
| "increase of output" | 3.46 | 3.52 | 0.40 |
| "stabilization of the economic situation of the family" | 4.34 | 4.33 | 0.24 |

The table includes respondents among whom a successor is known (42 cases) or not known (56). In the remaining cases (22), respondents chose the option "it is difficult to say whether a successor is known". N = 120; the hierarchy of each objective from 1 to 5; *p* *—refers to the significance of Mann-Whitney U test for two independent groups. Source: Own study based on the questionnaire survey.

The perspective of having a successor encourages paying more attention to economic objectives in the functioning of the agricultural holding. However, the lower interest of respondents in the case of the objective "care for the environment" is noteworthy. It is difficult to explain it clearly. Probably the lack of perspective of a successor lowers the interest in economic objectives, at the same time compensating for it with a greater interest in the environmental objective. This does not automatically mean pro-environmental investments but rather their perception. It turns out that in 50% of all surveyed households which carried out at least two activities related to pro-environmental investments in 2016–2019, a successor was known, while in 36% the successor was not known. Therefore, the perception of the respondents, in this case, is different from real action.

The relationships between most of the objectives of the functioning of the farms in the surveyed units turned out to be statistically significant, but their strength was not high and in many situations even low (Table 5).

The relation between the objectives "providing of income" and "increasing the value of assets" proved to be moderately weak (from the perspective of Spearman's rank correlation) also in comparison with the other objective. Thus, the first hypothesis could be only partly accepted. This means that despite the existence of quite obvious links between income and assets in the theoretical layer, in practice they are far more complex. The correlation coefficient between income and assets was statistically significant in the analyzed group (0.48) and after eliminating 5% of the outliers (0.55). Interestingly, the links between the objective of "care of the environment" and the other economic one show a weak strength, statistically insignificant, but positive. This confirms the research hypothesis. Taking care of the environment is additionally connected with the attitude of the agricultural producer as a consumer, and as a consequence, it is determined not only by his production function. As can be seen from the meta-analysis conducted by Dessart et al. [47], behavioral factors are also important here. When farmers have sufficient competence and knowledge of environmental practices, there are environmental and economic benefits associated with these limited risks and most neighboring farmers have done so, then the adoption of sustainable practices is higher. Moreover, there are differences between the respondents' perception of the importance of objectives and their actual implementation. This is related to the need to adapt to institutional conditions (meeting the requirements for receiving subsidies), as well as the fact that investments (e.g., in machinery, new crop technologies) increase productivity and are more and more eco-friendly. Therefore, the funds available under the CAP are important for investment incentives for farms [48], which is favorable to achieve both economic and environmental objectives. Moreover, it was noted that the economic objectives are relatively complementary to the social objectives in the group of the examined farms. This is also confirmed by another study [49] based on farms using graziers in Australia, which shows that income objectives were complementary with social objectives, but for individuals with a history of expansion.

At the next stage of the research, factors determining the choice for the most preferred objective, i.e., providing of income in the examined group of respondents, were identified. It concerns, among others, the identification of these factors as well as the discovery of rules related to them. An exploratory method of variable analysis was used—classification trees (C&RT). In the case of the objective of "providing of income", a tree consisting of five final nodes and four divisions was selected for further research (Figure 1). The choice of this tree resulted from the relatively low cost of the cross-check test as well as a reasonable level of resubstitution costs, which was also reflected in its automatic selection, as the optimal tree, by the Statistica program from among seven trees (the cost of the cross-check was 0.3 for the selected tree, the standard deviation of the cross-check was 0.045 and the resubstitution cost was 0.225). The selection was confirmed by a fairly good match with the data. It has correctly classified as many as 89% of the cases as "no" for the objective under consideration and 64% as "yes", i.e., approximately 78% of the correctly classified cases.

**Table 5.** Correlation of Spearman's rank between the objectives of the examined agricultural holdings in the Wielkopolska region.

| Specification | Providing of Income | Increasing the Value of Assets | Providing Funds for the Family | Modernization of Farm | Ensuring Continuity of Running | Care for the Environment | Increase of Output | Stabilization of the Economic Situation of the Family |
|---|---|---|---|---|---|---|---|---|
| providing of income | | | | | | | | |
| increasing the value of assets | 0.32 * | | | | | | | |
| providing funds to the family | 0.56 * | 0.32 * | | | | | | |
| modernization of the farm | 0.19 * | 0.25 * | 0.37 * | | | | | |
| ensuring continuity of running | 0.32 * | 0.35 * | 0.40 * | 0.47 * | | | | |
| care for the environment | 0.11 | 0.08 | 0.09 | 0.20 * | 0.20 * | | | |
| increase of output | 0.31 * | 0.33 * | 0.32 * | 0.42 * | 0.39 * | 0.20 * | | |
| stabilization of the economic situation of the family | 0.52 * | 0.27 * | 0.52 * | 0.23 * | 0.47 * | 0.22 * | 0.32 * | |

N = 120. * Statistically significant correlations of 0.05 are indicated. Source: Own study based on the questionnaire survey.

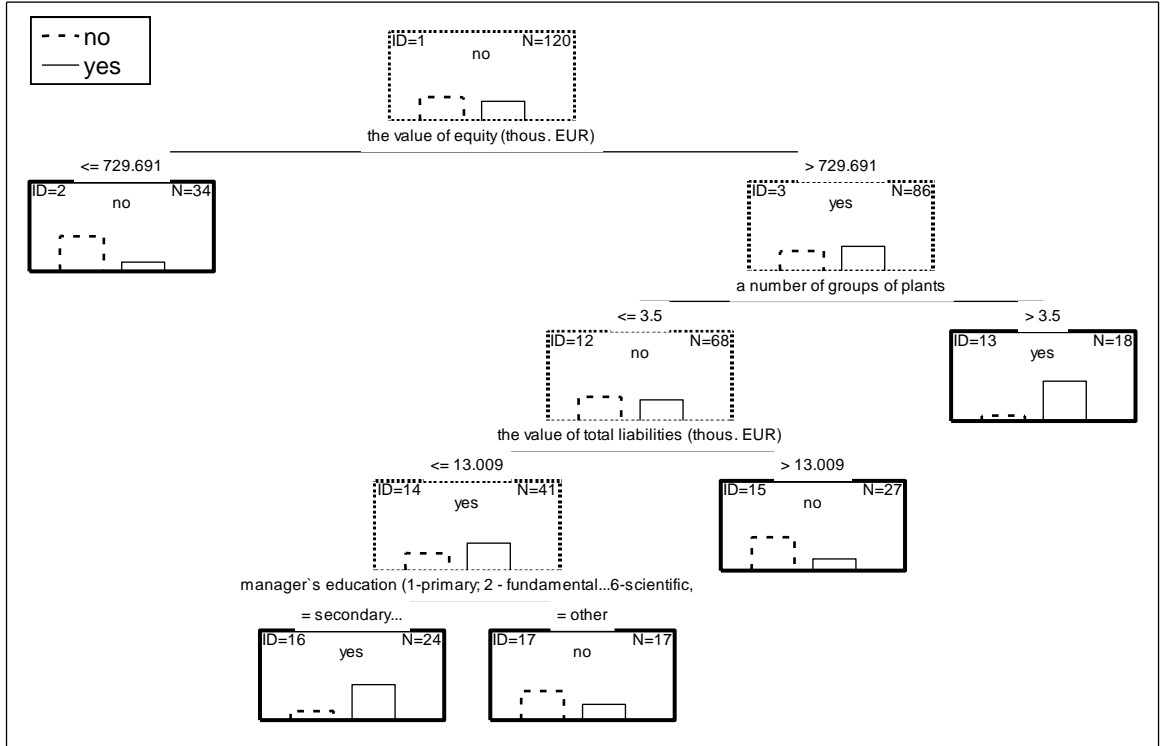

**Figure 1.** The model of the final classification tree (C&RT) in regards to the selection of the highest preference for the objective of "providing income" in the group of examined respondents (N = 120). Yes—means the maximum preference of the respondents for the income objective, No—other cases; ID means the node. We have five end nodes according to the analyzed model of the C&RT: ID1—yes = 55, no = 65; ID2—yes = 7, no = 27; ID13—yes = 16, no = 2; ID15—yes = 7, no = 20; ID16—yes = 19, no = 5; ID17—yes = 6, no = 11; the values are given in EUR thousand for the value of equity and liabilities. Source: Own study based on the questionnaire survey.

The analysis of the tree structure indicates that the first factor of division that had the greatest influence on the highest preference for the objective of "providing of income" was the value of equity (Figure 1). Based on this tree, rules can be formulated for the respondents' choice of the highest preference for the objective of "providing of income". It follows that the appropriate choice of preferences for the examined objective by respondents is complex and depends on many factors. The simplest rule of the choice we are interested in refers to node no. 13. These are farms where the value of equity exceeds EUR 170.5 thousand and simultaneously four or more groups of plants are grown on arable land. Growing so many groups of plants may result from the necessity to meet the conditions for receiving payments for greening in the CAP. For farms in Poland using more than 30 hectares of arable land, at least three crops are required, including the provision that the main crop cannot occupy more than 75% of arable land, and two main crops together cannot occupy more than 95% of arable land. This means that there is an integration of the economic (relatively high level of equity capital, which allows creating higher income) and environmental (several crops—biodiversity) dimensions. Growing so many groups of plants may initially indicate the environmental sustainability of these farms in the field of biodiversity. Achieving a certain level of own capital does not guarantee that the producer will prefer to increase the agricultural income very highly. Other factors are also important. When the value of total liabilities is less than 3000 EUR and the farm manager has a secondary education, it also favors preferences for the respondent's choice. It is worth noting here that such a set of features concerns the most numerous group of respondents opting for the maximum preference in the case of the objective "providing of income". This indicates a complex nature of the

selection of preferences with regard to objectives and thus directions of development at the level of a farm.

## 5. Discussion

The researches indicate that the surveyed farmers in their hierarchy had the highest perception of the income objective. The open question is whether the perception of income is really an aim or a means to achieve other objectives? The results of the analyses, especially multidimensional scaling and decision trees, allow concluding that this does not contradict each other. This is also confirmed by the results of other studies [50], which show that income (profit) in the sense of maximization can be both an objective and a category facilitating the achievement of other objectives.

The lack of stronger links between the income and asset objective, both in terms of the perception of objectives and economic factors (from the perspective of Pearson's correlation coefficient) results from the fact that land as the main component of assets shows some peculiarities. It is a matter of separating its prices from its productivity. This is related to new non-productive uses of the land, a speculative motive. It is also important to capitalize subsidies in the price of land. This is particularly important in the new EU member states where there are no so-called historical area payment entitlements [51]. In addition, the assets are seen in terms of current choices to a lesser extent than income.

Similar results were obtained by Harrison and O'Brien [52] by studying dairy farms in New South Wales (Australia). They concluded that economic objectives were more preferable to the objective "conservation of natural resources". In turn, a study [53] conducted on 257 dairy farms in the Netherlands shows that the most important objective for farm users was job satisfaction and the production of quality and safe products. Meanwhile, income maximization was only in fifth place. These results may indicate that these farmers consider the internal value of agriculture to be more important than the economic value. This may be due to the relatively good economic situation of Dutch dairy farmers, which allows them to prefer other objectives than income maximization.

In view of the above, it cannot be ruled out that the increase in the prosperity of agricultural producers may reevaluate the hierarchy of objectives. Thus, it is only by reaching a certain income threshold and operating in an environment that affirms certain values making farmers value non-economic objectives more, as is the case with the environmental concept of the Kuznets curve. As the income level of farms in Poland is still lower than in the Netherlands, the income objective remains the most important. Moreover, the results of a study by Kallas et al. [17] carried out on Catalonian vineyard farmers confirm the highest preference for the maximized farm income objective, interestingly also for organic farms. In contrast, for farmers in a medium-development country, i.e., Turkey, the highest ranking in farmers' objectives was for risk minimization, followed by profit maximization [54]. This confirms previous conclusions about the experience from Poland in the 1990s, when the issues of risk minimization were strongly exposed by farmers [18]. It can be expected that together with economic development, the income objective, but also the environmental objective, will be more preferred. Moreover, the referred studies as well as analysis related to the classification and regression trees indicate that the choice of the objectives by agricultural producers is a complex mechanism. As Amador et al. [55] showed, based on research in Andalusia, the farmer attempts to achieve several objectives, most of which are in conflict.

It is interesting that, as conducted research shows, as the economic size of farms increased, farmers rated the income and asset objective higher, with a slightly lower environmental one. For example, Ripoll-Bosch et al. [56] underlined that small farms, those with low income and assets, are more environmentally friendly due to, for example, lower environmental pressure. Moreover, Briner et al. [57] studying interactions between the economic and environmental dimension, on the example of mountain regions in Switzerland, pointed to a trade-off in the case of economic outcome and water use. Hence, there is a conflict between economic and environmental objectives. On the other hand, Jan et al. [58] on the base of the research of Swiss dairy farms, provides evidence that there is no trade-off between economic and global environmental farm performance. In turn, as Mutyasira et al. [59]

noted, larger farms are better at implementing environmentally-friendly practices. Although these producers do not position the environmental objective relatively high, they have larger capital resources that enable them to implement pro-environmental investments, which are often conducive to improving economic efficiency (lower energy intensity of production). Therefore, farmers managing larger farms were more likely to take pro-environmental measures. In many areas, larger farms are obliged to take such actions to receive subsidies under the CAP rules (cross-compliance rules or those related to receiving payments for greening). Moreover, the adoption and promotion of best farming techniques, eco-innovation, or services that require income, capital, and are associated with environmental performance improvement [10,60]. This would mean that there is complementarity and not substitutability between these fields. As stressed by Beltrán-Esteve and Picazo-Tadeo [61], environmental policies aimed at boosting catching-up are highly recommended, especially in the newer member states of the EU.

As for the factors shaping the highest preferences for the income objective in the group of the surveyed respondents (Figure 1), it is worth noting that we have linked with the economic dimension (value of equity, liabilities) and also the environmental dimension (number groups of a plant). These relationships are real and not resulting from the perception of the processes taking place in the farm. This would confirm other results of research on positive relations between these dimensions in agricultural holdings [62]. Therefore, subsidizing the use of green practices in agriculture is required [63]. This is mainly about the further development of a system of positive incentives that would motivate farmers to take more pro-environmental measures due to the fact that they do not see the environmental objective high in the hierarchy. Other studies also stress that the interaction between farmers, advisors, and experts plays a central role in shaping sustainability in farms, especially between the economic and environmental dimensions [64]. Among other things, it is emphasized that the implementation of Good Agricultural Practices, those that favor the environment, should focus mainly on low-income farmers [65].

## 6. Conclusions

The conducted research confirmed in part the first hypothesis (there is a strong link between the choice of income and asset objectives by respondents). Thus, the relationship between the income and assets objective proved to be moderate in terms of strength. This may be related to the functioning of the land market and the consequent decoupling of the prices of this main asset from productivity. Moreover, in the perception of the asset in terms of the long term (also in less direct terms) rather than the current moment. Research has shown a very weak positive link (statistically insignificant) between economic and environmental objectives, which support the second hypothesis (there is not a clear, but positive link between the economic objectives (providing income, increasing the value of assets) and the environmental objective). As it appears from the analyses of multidimensional scaling, the environmental objective is out of line with the others, especially in relation to the income. Therefore, it is difficult to state unequivocally whether there is a complementarity between the income and the environmental objective from the perspective of perception by respondents. Although when the context of the real action is taken into account then the answer should be positive.

There are, as the results of the cluster analysis indicate, differences in the field of the positioning of the objectives, including also the group of respondents in which the income and environmental objectives have been ranked high at the same time. It is worth stressing that this is about the relationship of objectives in the context of their perception by respondents. The introduction of institutional solutions such as cross-compliance rules in the CAP, meeting the conditions for receiving payments for greening, payments for the implementation of higher environmental standards—agri-environmental payments, allows mitigating possible antagonisms between them based on real action. This makes farms with greater economic strength more often undertake pro-environmental measures. The managers of these farms were obliged to undertake such measures under institutional pressure. Moreover, they had more funds at their disposal for this type of investment, which to a large extent are environmental-friendly

in nature and have an impact on reducing the energy intensity of production. It is quite a new topic in the scope of the divergence between the perception of the objectives and real operation, which should be developed in the future.

The most important objective of the functioning of the examined farms is "providing of income". This objective was the highest rated in terms of preferences among the respondents. On the other hand, however, the objectives such as: "Stabilization of the economic situation of the family" and "providing funds for the family", also achieved high (albeit lower) marks. It may be assumed that small differences between these objectives resulted more from the behavioral characteristics of farm managers, their preferences in relation to family values, the engagement in family life, or treating the income objective only as a means of achieving other objectives [66]. It is worth stressing that, on average, the environmental objective received a slightly higher preference than the second in the hierarchy of economic objectives, i.e., increasing the value of assets, as well as other economic objectives. This may result from the existing pro-environmental trend in agricultural policy in the European Union, also from the growing awareness of agricultural producers, as well as the social pressure related to it. In larger units, the income objective, also the asset objective (although to a lesser extent), and the environmental objective were even rated higher by respondents. However, as it results from these surveys, larger farms more often took pro-environmental measures. The perspective of having a successor made the evaluations for the economic objectives, including in particular the objectives "increasing the value of assets" grow. Thus, ensuring succession favors economizing the functioning of an agricultural holding.

It was noticed that the factors that influenced the choice of the highest preference for the income objective (while at the same time lower for the asset objective) was the value of equity capital exceeding EUR 170 thousand, also a number of groups of plants grown on arable land (more than three), as well as the level of debt and education. This means that there is a complex mechanism shaping the preferences of respondents in regards to the income objectives, and the selection of preferences is related to both the economic and the environmental factors. The results of the research indicate initially the possibility of a gap between the farmers' perception of reality (choice of the hierarchy of objectives) and real actions. It is a new issue which points to the need to stimulate the environmental objective in particular through support at the level of agricultural policy instruments.

Further research on the objectives of agricultural holdings should be continued due to the dynamic nature of changes in the economic, social, and institutional environment. It also concerns the verification of directions and effectiveness of agricultural policy both at the level of the Member States and the EU. It would be particularly interesting, from the scientific point-of-view, to repeat these studies in the same research group every few years. In light of the presented research results, both the objectives and instruments of agricultural policy should also influence the demographic and social features of farmers' households. The aim is to create favorable conditions for the successors of agricultural holdings, as well as the further multifunctional development of rural areas, which enables diversification of income and better satisfaction of social needs, including cultural and educational ones. It is also important to increase diversification and better identify the needs of farmers in the context of the work carried out by agricultural advisors. In this way, advisory services can be used more effectively. In addition, there is still a challenge to further improve incentives for the non-antagonistic coexistence of economic and environmental objectives both in terms of perception and implementation. In the future, it is worth considering raising the production standards for receiving subsidies to larger farms, especially those with a high livestock density. On the other hand, for small farms that generate relatively a low income, the creation of a system of green investment grants could be considered. This is also important because, as indicated by Guth et al. [67], the distribution of support under the CAP favored the largest farms, increasing disparities within the sector. It is about stimulating sustainable development taking into account both the economic objectives of agricultural producers (income) and environmental objectives.

**Supplementary Materials:** The following are available online at http://www.mdpi.com/2077-0472/10/10/458/s1. The questionnaire S1 (selected questions used in the article); Table S1: List of the explanatory variables used in the classification tree model; Table S2: The coordinates for specific objectives in the agricultural holdings surveyed in the Wielkopolska region in multidimensional scaling; Figure S1: The distribution of the objectives of operating in the examined agricultural holdings in Wielkopolska (N = 120) in a two-dimensional space from the perspective of the multidimensional scaling; Figure S2: The dendogram (the cluster analysis—the agglomeration) of the examined agricultural holdings in the Wielkopolska region due to the objectives of their operation; Table S3: The mean of the objectives of operation of the examined agricultural holdings in the Wielkopolska region due to the distinguished clusters.

**Funding:** This article is founded by the National Science Centre in Poland (grant no. 2018/29/B/HS4/01844).

**Acknowledgments:** I thank Michał Borychowski and Jakub Staniszewski from Poznań University of Economics and Business for help in the questionnaire.

**Conflicts of Interest:** The author declares no conflict of interest.

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
