# Peer review of "The Objectives of Farm Operations—Evidence from a Region in Poland"

_agriculture, doi:10.3390/agriculture10100458_

Round 1
Reviewer 1 Report
The objectives of farm operations – evidence from a region in Poland
The aim of the paper is to recognize the preferences of farm managers in the scope of the objectives of farms' operations, and also to examine the mutual relations between those objectives.
In my opinion, the subject studied in this work is of great interest both for farmers and for the design of agricultural policy. The work is well written and presents an interesting discussion of the results. However, I think that the authors should improve some aspects, such as those that are exposed next.
- The 120 farms in the sample should be described in more detail, taking into account their size and productive orientation.
- Line 158 indicates that the farms have been selected according to their size, and it seems to indicate that they correspond to group ES6. However, line 169 indicates that the sample is distributed among groups E2-E5. This contradiction must be clarified.
- The variables used must be clearly described, indicating those that come from the survey and those that are extracted from the FADN.
- It would be interesting to attach the survey (as an Annex).
- In Table 2, the titles "mean group 1" and "mean group 2" are confusing. This should be improved.
- It would be interesting to present and analyze a table similar to table 3 but taking into account the productive orientation of the farms. In these tables it would be important to identify the statistically significant differences. Line 382 talks about clear differences in the economic objectives in relation to table 4: statistically significant differences should be shown.
- Footnote 1, as well in the text, talks about pro-environmental measures specified in the survey. I think these measures should be made explicit and explained.
- The results of the C&RT analysis should be reviewed: there are discrepancies between what appears in the text and what is observed in Figure 1. For example: the value of equity (line 448), value of total liabilities (line 466).
- The meaning of "yes" and "no" should be clarified in figure 1.
- Looking for some robustness in the results of the C&RT analysis, it would be interesting to carry out the analysis with other economic objectives.
Formal questions should be carefully reviewed. For example:
- I think line 12 (fourth line of the abstract) repeats "and still".
- In several lines of the text there is a "." when it should be a ",". For example, on lines 350, 378, 451, 502, 555, 581, ...
Reviewer 2 Report
- Page 1 line 5: duplicated words "and still and still."
- page 1 line 18-19 "while the environmental objective turned out to be poorly, positive related to economic objectives (in the context of their perception by respondents)" I am not sure what does the author mean by "poorly" but "positively"? "poorly" normally means statistically insignificant or unimportant. Please check the meaning.
- It is rare to see a figure in the abstract; it is better to separate and move the figure to the materials and methods/results and discussion section to summary the approaches taken/and the findings.
- Lines 66-68, I am not sure why the sentence is there : "Without the latter, it would be difficult to meet generational objectives when in the examined group the average share of income from agriculture in the total household income of a farmer's family is dominated and was 76%." Please explain.
- Lines 300-303 please revise the sentence to make sense.
- Please check the consistency in the unit expression for "," and "." for the thousand expression and the decimal point; and instance, note 2 under line 457 and the content of figure 1 (<=3,5 and >3,5); and also line 466 and at the axises of the figure S2.
- Between lines 404-405 (contents of table 5): as the correlation coefficients are symmetric, it is advised to keep only the low half of the triangle.
- The monetary unit should be used consistently; in most of the contents "EURO" was used, but in limited occasions the it was expressed as "EUR" (e.g., lines 448; 463, 586) or the Euro sign (line 466).
- There is an extra "." in the last line of Table S2; and in line 451 (it should be a "," instead).
- 10.There is a mis-spelling for ref[1]; the textbook title should be "microeconomics" not "macroeconomics."
- 11.There are too many parentheses used in the text to explain the restriction or definition used. The author might want to rewrite some of the sentences to enhance readability.
Reviewer 3 Report
Making use of cluster analysis, the submitted manuscript aims to evaluate the objectives of agricultural operations, focusing on a regional case in Poland. The purpose of the paper is interesting and might be of fit with Agriculture readership. Though it needs some major formal and substantial amendments. These would also help to make the work easier to understand.
First of all, the paper needs a thorough linguistic revision by a professional mother-tongue proofreader. As far as the abstract is concerned, the wording of the question in line 10-11 is not clear. Moreover, I would discourage the author from including a graph in that paragraph - it would be more suitable in the introduction.
In the introduction, I suggest removing the paper structure (lines 89-90) since it is eventually repeated in the introduction. Throughout the paper, the first person is used several times by the author. I recommend to always use an impersonal form.
Although the reviewed literature is appropriate, I trust that those references should be corroborated by additional publications on environmental, social and economic sustainability.
In the methodology, I suggest clarifying the structure of the questionnaire, perhaps reporting it in an appendix. The author could insert a graph that explains all the steps used to evaluate the questionnaire, the various analyses, the objectives and expected results.
The conclusions are consistent with the objectives of the paper, but I suggest to structure it in a more convincing way. For example, line 543 ("The conducted research confirmed in part the first hypothesis") I suggest to explain this hypothesis and put in brackets referring to the first hypothesis.
I recommend mentioning this work with regard to sustainability in agricultural areas and the methodology used.
- Agovino, M., Cerciello, M., & Gatto, A. (2018). Policy efficiency in the field of food sustainability. The adjusted food agriculture and nutrition index. Journal of environmental management, 218, 220-233.
- Rusciano, V., Civero, G., & Scarpato, D. (2017). Urban gardening as a new frontier of wellness: Case studies from the city of Naples. The International Journal of Sustainability in Economic, Social, and Cultural Context, 13(2), 39-49.
- Scarpato, D., Civero, G., Rusciano, V., & Risitano, M. Sustainable strategies and corporate social responsibility in the Italian fisheries companies. Corporate Social Responsibility and Environmental Management.
- Drago, C., & Gatto, A. (2018, March). A robust approach to composite indicators exploiting interval data: The interval-valued global gender gap index (IGGGI). In IPAZIA Workshop on Gender Issues (pp. 103-114). Springer, Cham.
- Rusciano, V., Civero, G., & Scarpato, D. (2020). Social and Ecological High Influential Factors in Community Gardens Innovation: An Empirical Survey in Italy. Sustainability, 12(11), 4651.
- Gatto, A. (2020). A pluralistic approach to economic and business sustainability: A critical meta‐synthesis of foundations, metrics, and evidence of human and local development. Corporate Social Responsibility and Environmental Management.
Round 2
Reviewer 1 Report
The authors did all revisions I asked of them.
Best regards,
Author Response
Thank you for appreciating the improvement of the article and friendly words
Reviewer 3 Report
1)The linguistic review has been almost sufficiently improved as well as the understanding of the paper - it now looks decisively better. It is worthwhile operating further linguistic polish.
2)The abstract is now clearer and the lines (10-11) are now understandable. As far as the graph is concerned, its deletion is the most correct choice.
3) The changes to the introduction are consistent with the requirements and the structure of the paper is more clarified
4) It is recommended to better insert the work within the existing literature, making use of additional key publications in the field. The suggested references to literature are well integrated into the text.
5)The methodological structure is well articulated and in conformity with the paper's objectives.
6)The changes to the conclusions both in content and language are almost enough.
Author Response
Poznań 1.10.2020
Response to reviewer
Dear Reviewer,
I would like to thank You kindly for the additional remarks that I have implemented in the article and your appreciation for the improvement of the article. I have applied the following changes, according to the suggestions:
- The linguistic review has been almost sufficiently improved as well as the understanding of the paper - it now looks decisively better. It is worthwhile operating further linguistic polish.
The English language was polished. This is mainly concerned with the improvement of grammar, and text intelligibility.
- It is recommended to better insert the work within the existing literature, making use of additional key publications in the field. The suggested references to literature are well integrated into the text.
The additional literature was employed which, in my view, additionally enriches the considerations:
- Klasen, S.; Meyer, K. M.; Dislich, C.; Euler, M.; Faust, H.; Gatto, M.; Hettig; E., Melati, D.N.; Jaya, N.S.; Otten, F.; Pérez-Cruzado, C.; Steinebach, S.; Tarigan, S.; Wiegand, K. Economic and ecological trade-offs of agricultural specialization at different spatial scales. Ecol. Econ. 2016, 122, 111-120. https://doi.org/10.1016/j.ecolecon.2016.01.001 (77-78, footnote 1: 392-393)
…It is particularly important whether there is a conflict between the economic and environmental objectives on farms [7, 8]
…It is worth pointing out, however, that the problem of the importance of specialization for economic and ecological trade-offs of agricultural specialization has been addressed, for example by Klasen et. al. [8].
- Jan, P.; Dux, D.; Lips, M.; Alig, M.; Dumondel, M. On the link between economic and environmental performance of Swiss dairy farms of the alpine area. Int. J. Life Cycle Assess. 2012, 17, 706–719. https://doi.org/10.1007/s11367-012-0405-z. (570-572)
…On the other hand, Jan et al.[58] on the base of the research of Swiss dairy farms, provides evidence that there is no trade-off between economic and global environmental farm performance.
- Beltrán-Esteve, M.; Picazo-Tadeo, A.J. Assessing environmental performance in the European Union: Eco-innovation versus catching-up. Energ. Policy 2017, 104, 240-252. https://doi.org/10.1016/j.enpol.2017.01.054 (583-585)
… As by stressed Beltrán-Esteve and Picazo-Tadeo [61], environmental policies aimed at boosting catching-up are highly recommended, especially in the newer member states of the EU.
- Besides, the introduction (it was mainly about the background of considerations) has been improved by adding two sentences (87-91):
….It should also be noted that farmers' perceptions of the hierarchy of the objectives of farm operations may not be adequate for the activities they undertake. Therefore, the recognition of these processes may contribute to more effective implementation of pro-environmental measures by farmers (problems of economic incentives, education, improvement of awareness
- The description of the methods has been improved by adding two sentences (202-205)
…This made it possible to obtain research material of high reliability. Only in a few cases (9) did the questionnaires require supplementing or individual explanations from the interviewers, also in situations of so-called outlier observations.
- Also, the narrative in the discussion has been improved towards a more homogeneous communication (564-587)
… It is interesting that, as conducted research shows, as the economic size of farms increased, farmers rated the income and asset objective higher, with a slightly lower environmental one. For example, Ripoll-Bosch et al. [56] underline, that small farms, so those with low income and assets, are more environmentally friendly due to, for example, lower environmental pressure. Also, Briner et al. [57], studying interactions between economic and environmental dimension, on the example of mountain regions in Switzerland, point to trade-off in the case of economic outcome and water use. Hence, there is a conflict between economic and environmental objectives. On the other hand, Jan et al.[58] on the base of the research of Swiss dairy farms, provides evidence that there is no trade-off between economic and global environmental farm performance. In turn, as Mutyasira et al. [59] note, larger farms are better at implementing environmentally- friendly practices. Although these producers do not position the environmental objective relatively high, they have larger capital resources that enable them to implement pro-environmental investments, which are often conducive to improving economic efficiency (lower energy intensity of production). So farmers managing larger farms were more likely to take pro-environmental measures. In many areas, larger farms are obliged to take such actions to receive subsidies under the CAP rules (cross-compliance rules or those related to receiving payments for greening). Also, the adoption and promotion of best farming techniques, eco-innovation, or services that require income, capital and are associated with environmental performance improvement [10, 60]. This would mean that there is complementarity and not substitutability between these fields. As stressed by Beltrán-Esteve and Picazo-Tadeo [61], environmental policies aimed at boosting catching-up are highly recommended, especially in the newer member states of the EU.
Once again, I would like to thank you for all the valuable comments.
Sincerely,
Aleksander Grzelak PhD
(corresponding author)
